# Recent Advances in Two-Dimensional MXene-Based Electrochemical Biosensors for Sweat Analysis

**DOI:** 10.3390/molecules28124617

**Published:** 2023-06-07

**Authors:** Selvaganapathy Ganesan, Kalaipriya Ramajayam, Thangavelu Kokulnathan, Arunkumar Palaniappan

**Affiliations:** 1Department of Chemistry, School of Advanced Sciences, Vellore Institute of Technology, Vellore 632014, Tamil Nadu, India; selvaganapathy.g@vit.ac.in (S.G.); kalaipriya.r@vit.ac.in (K.R.); 2Centre for Biomaterials, Cellular and Molecular Theranostics, Vellore Institute of Technology, Vellore 632014, Tamil Nadu, India; 3Department of Electro-Optical Engineering, National Taipei University of Technology, Taipei 106, Taiwan

**Keywords:** point-of-care, diagnostic, wearable sensor, glucose monitoring

## Abstract

Sweat, a biofluid secreted naturally from the eccrine glands of the human body, is rich in several electrolytes, metabolites, biomolecules, and even xenobiotics that enter the body through other means. Recent studies indicate a high correlation between the analytes’ concentrations in the sweat and the blood, opening up sweat as a medium for disease diagnosis and other general health monitoring applications. However, low concentration of analytes in sweat is a significant limitation, requiring high-performing sensors for this application. Electrochemical sensors, due to their high sensitivity, low cost, and miniaturization, play a crucial role in realizing the potential of sweat as a key sensing medium. MXenes, recently developed anisotropic two-dimensional atomic-layered nanomaterials composed of early transition metal carbides or nitrides, are currently being explored as a material of choice for electrochemical sensors. Their large surface area, tunable electrical properties, excellent mechanical strength, good dispersibility, and biocompatibility make them attractive for bio-electrochemical sensing platforms. This review presents the recent progress made in MXene-based bio-electrochemical sensors such as wearable, implantable, and microfluidic sensors and their applications in disease diagnosis and developing point-of-care sensing platforms. Finally, the paper discusses the challenges and limitations of MXenes as a material of choice in bio-electrochemical sensors and future perspectives on this exciting material for sweat-sensing applications.

## 1. Introduction

The current healthcare system is taking a transition from a conventional approach, where patients visit a healthcare professional for any medical aid after developing the disease or its symptoms, towards more sophisticated continuous monitoring of vital biomarkers and then informing patients (sometimes by healthcare professionals remotely) of any deviation from the normal level. This enables early detection of any deviation from the level of the normal biomarker and brings in new possibilities for the prevention of disease that are not possible through the conventional system. This requires highly specific and sensitive biosensing platforms for precisely and continuously monitoring the biomarkers. The electrochemical-based wearable sensors are currently being explored on this line and are found to be an excellent sensing platform to address the above challenges [1,2,3]. Thus, wearable electrochemical biosensors can provide continuous, instantaneous data on patients’ physiologies via dynamic, non-invasive measures of biomarkers in biofluids without causing discomfort to the wearing individuals [4,5,6,7,8]. In recent years, the growth of wearable electronic technologies that can precisely quantify vital responses such as body temperature, blood pressure, blood sugar level, and heart rate, aiding in depicting and monitoring the individual’s health conditions, has accelerated [3,9]. However, these biological parameters cannot provide complete information on the human body’s robust biochemical and metabolic functions [10]. So, biofluids such as tears, saliva, and sweat are of massive interest to biosensors researchers for their ease of sampling and their demonstrated capabilities to provide continuous and instantaneous monitoring of vital biomarkers and parameters, which could give insights into the subject’s physiology in a non-invasive manner [11,12,13,14,15,16]. Compared to other biofluids, sweat is high in several essential analytes such as glucose [17], lactate [18], cortisol [18], uric acid [19], interleukin [20], ammonium [21], sodium [22], potassium [23], calcium [24], iron [25], and zinc [26]. They provide critical physiological information about the human body and are closely intertwined with capillaries and nerve fibers, offering an immense advantage for biosensing applications. Moreover, wearable analytes monitoring technologies at sweat production sites open a huge possibility of autonomous, continuous, and instantaneous sensing of vital analytes present in sweat.

The first work on wearable electrochemical sensors for real-time monitoring of sweat lactate was developed by Jia et al. [27]. Further, many researchers have executed the monitoring of metabolites, electrolytes, drugs, and trace elements present in sweat. Electrochemical sensing is a familiar and deep-rooted method for detecting sweat analytes that is extensively used in wearable sensing platforms and often used in clinical diagnostics due to its simplicity, portability, high performance, and low cost [28,29,30]. Some recent technological developments are steering the sophistication of wearable sweat sensors based on electrochemical methods. A complete unified multiplexed sensing device for simultaneous monitoring of multiple analytes makes the system more practical. It offers a versatile wearable electrochemical biosensing platform for large-scale clinical diagnostics and physiological understandings [31,32,33]. Thus, the performance and effectiveness of the wearable biosensor rely heavily on the properties of materials used to make the device.

Some of the most common materials used in biosensors include and are not limited to the following: graphene and its derivatives [34,35], transition metal nanoparticles [36,37,38,39], transitional metal dichalcogenides [40,41], gold nanoparticles [42,43], silver nanoparticles [44,45], and very recently, transition metal carbides/nitrides, called as MXenes [46]. Because of their excellent multifaceted characteristics including high surface area, size control, tunable electronic and mechanical properties, high biocompatibility, exceptional sensitivity, specificity, and low detection limit (LOD), MXenes are currently being explored heavily for fabricating biosensors compared to other nanomaterials. MXenes are a class of 2D inorganic nanomaterials with few-atom thicknesses made of layered transition carbides, nitrides, and carbonitrides [47,48,49]. They are typically exfoliated by etching of the A layer from the MAX phase, specifically [50]. The general formula of an MXene is M_n+1_X_n_T_x_, where M represents the early transition metals, X is for carbon or nitrogen, T is for surface functional groups (–OH, –F, =O), and n represents an integer.

This review systematically summarizes recent breakthroughs in MXene-based electrochemical biosensors for sweat analytes. We review and discuss various synthesis methods for MXenes, their surface chemistry, and functionalization strategies in sensor applications. The advantages and importance of sweat analytes in continuously monitoring human physiology and disease conditions are also highlighted. Finally, we also discuss various MXene-based wearable biosensors for vital analytes in sweat. Most importantly, this review sheds light on future scopes and recommendations for researchers working on MXenes for electrochemical biosensor applications.

## 2. Advantages of Sweat as an Analyte in Biosensing

In the current healthcare systems, blood serum is considered as the gold standard to determine the concentration of analytes in most cases [51,52]. However, the invasive methods of sampling blood from patients produce more impediments, particularly for hemophobic patients, new-borns, and elderly persons. Urine is also a biofluid widely used as clinical samples but has some limitations for autonomous and continuous monitoring [53]. Saliva is another biofluid that contains many biomarkers, including enzymes, hormones, antibodies, and antimicrobial agents, which can precisely provide details on the molecular state of human physiology [54]. However, saliva monitoring also has specific drawbacks, such as saliva in the mouth containing many impurities and food particles, which could affect the precision of results. Tears, another biofluid, contain enzymes, proteins, lipids, and certain salts, which can reflect the conditions and disease in the eyes and others [55]. Unfortunately, the present tear sample collection protocols produce reflex tears and eye irritation, which can also affect the results of sensor analysis.

In contrast to other biofluids such as blood, urine, saliva, and tears, sweat has a massive advantage in biosensing applications. Sweat regulates heat balance in the body and plays crucial physiological roles including immune defense, thermoregulation, moisturization, and pH balance [56,57]. Thus, sweat contains several biomarkers that can provide the complete physiological status of the body at the molecular level [58,59]. The sweat glands in the body secrete the sweat. Thus, sweat analytes can be collected in a non-invasive manner from different body parts and are optimal for continuous monitoring. The average density of sweat glands in the body and the approximate range of analytes’ concentrations in the sweat fluid are shown in Figure 1a,b [60]. Key analytes in human sweat and associated health conditions are shown in Table 1.

## 3. MXenes–A Material of Choice for Biosensors

In contrast to metal nanoparticles- and graphene derivative-based biosensors, MXenes have gained much attention in developing biosensors due to their excellent biocompatibility and high electrical conductivity properties. A suitable sensor possesses high specificity, sensitivity, LOD, quick response, and a wide operating range. In addition, it also needs to be less expensive not only in lab-scale use, but also while scaling up. MXenes have all/most of these very important properties in the development of biosensors [81,82,83,84].

### 3.1. Synthesis Strategies of MXenes

MXenes, a new class of 2D anisotropic nanomaterial, have significantly improved from their first discovery in 2011 (Ti_3_C_2_T_x_) by the selective etching of the MAX phase precursor Ti_3_AlC_2_ [85]. Typically, MXenes are developed by removing the A layer from their MAX phase via specific etching. The synthetic methods of MXenes can be classified into two types: the top-down approach and the bottom-up approach. The top-down approach involves the direct exfoliation of the A layer, while the bottom-up approach is based on 2D ordered growth from atoms/molecules. Both the methods are discussed in detail in the below sections.

#### 3.1.1. Top–Down Approach

The top–down approach involves the direct exfoliation of the bulk parent MAX phase material with its original integrity retained. These exfoliation processes are typically carried out using chemical and mechanical methods. The top-down approach can be further classified based on the type of precursors used, the delamination process, and the etchants involved in the reaction. The following section describes these methods in more detail.

##### Based on Precursors

Based on the precursors selected, the preparation strategy of MXenes is classified into MAX-phase and non-MAX-phase methods. In the MAX phase-based preparation method, the A layer is eliminated by using specific selective etchants with optimized concentrations and time intervals, followed by filtration and sonication to obtain few-layered 2D nanosheets. A typical example of this method is the elimination of the A layer from the MAX phase (Ti_3_AlC_2_) to form Ti_3_C_2_T_x_ MXene by Naguib et al. (Figure 2a), with hydrofluoric acid (HF) as an etchant in room temperature conditions [85].

Non-MAX phase-based synthesis was obtained from the selective etching of the gallium (Ga) layer from the Mo_2_Ga_2_C with 50% concentrated HF acid to form Mo_2_CT_x_ MXene [86]. In contrast to the known MAX phase formula, the non-MAX phase method of prepared Mo_2_CT_x_ MXene consists of two A layers of Ga stacked between the Mo_2_C layers. To confirm perfectly etched treatment, the X-ray diffraction patterns of Mo_2_CT_x_ and Mo_2_Ga_2_C before and after etching were compared (Figure 2b), in which there was a significant reduction in the peak intensity of Mo_2_C, which confirms the Ga phase was dissolved during the etching process.

##### Based on Delamination

In general, synthesizing single-layered MXenes requires further processing steps on the multi-layered MXenes, called delamination. The delamination of multi-layered MXenes can be by the intercalation method or through mechanical exfoliation. It was reported that mechanical exfoliation is ineffective because the interlayer interactions in MXenes produced by this method are highly tacky (the interlayer interaction in the MXene is two- to six-folds stronger than that present in graphite and bulk MoS_2_ [87]). Moreover, the sonication time also affects the lamellar structure, decreases MXene sheet size, and produces defects in the structure [88,89,90].

In the case of the intercalation method, intercalants are introduced to reduce the interlayer spacing, weaken the interaction between the MXene layers, and facilitate the formation of individual nanosheets. This dramatically improves the surface area and abundant surface terminations directly associated with the MXene’s electrically conductive properties. The intercalants widely used for intercalations of the MXene are classified into two types: ionic aqueous solutions that include metal hydroxides or halide salts as aqueous solutions [91,92] and organic intercalants such as dimethyl sulfoxide (DMSO) [93], tetrabutylammonium hydroxide (TBAOH) [94], tetra propylammonium hydroxide (TPAOH) [95], isopropyl amine [96], n-butyllithium [97], and bovine serum albumin (BSA) [98].

##### Based on Etchants

HF etching

In the HF etching-based MXene synthesis method, the layered MAX phase is stirred with HF aqueous solution at room temperature with specified concentration and time. Here, the multi-layered MXene is produced by an etched A layer from the MAX phase, and M-A bonds are replaced by the weak intercalations of T_x_ termination such as (–OH, –F, =O) on the surface of the multi-layered MXene. In agreement with numerous studies in recent times, several etching parameters such as temperature, time, and concentration of etchant play a deterministic role in the standard of the prepared MXene nanosheets. Kumar et al. demonstrated the effectiveness of Ti_3_C_2_T_x_ MXene etched at elevated temperatures for a binder-free supercapacitor application (Figure 3a–d) [99].

Non–HF etching

Even though HF is extensively and effectively utilized to prepare MXenes from the MAX phase, it is highly corrosive and hazardous to humans and the environment [100,101,102]. A trace amount of HF remaining in the MXene sample will harm the composite preparation and biological studies carried out further. Thus, the etching of MXenes by non-HF-based solvents is also extensively studied. The alternative etchants used for etching MXenes are weak acids and environmentally friendly bifluorides such as sodium bifluoride (NaHF_2_), ammonium bifluoride (NH_4_HF_2_)_,_ and potassium bifluoride (KHF_2_). Feng et al. demonstrated the synthesis of Ti_3_C_2_T_x_ MXene from the Ti_3_AlC_2_ MAX phase with bi-fluoride in a single-stage process. They found that the NH_4_^+^, Na^+^, or K^+^ ions enter the interlayer spacing of the MXene and enlarge the interplanar spacing and delamination efficiency [103].

In comparison to carbide-based MXenes, the Al layer is strongly bonded in the nitride-based MXene. Thus, more energy is required to remove the A layer from the nitride-based MXene. Moreover, the nitride-based MXene is less stable and might be able to dissolve in HF [104,105]. To overcome this issue, molten fluoride is used with the support of high-temperature heating to remove the A layer from the nitride-based MXene. Urbankowski et al. heated the Ti_4_AlN_3_ powder at 550 degrees for 30 min under an argon atmosphere with molten fluoride, where the free F^−^ ions were active enough to etch the Al layer from the nitride MXene MAX phases [106]. Studies revealed that the etchants are an essential determinant in the surface termination of the synthesized MXene. The fluorine-based etchants increase the percentage of F content in the surface terminations. Therefore, further modifications are needed for particular applications such as biomedical and sensors [107,108].

#### 3.1.2. Bottom–Up Approach

MXenes can also be synthesized by crystal growth methods using a small organic or inorganic molecule as the precursor. The bottom-up approaches enable precise manipulation of the size of the MXene sheet, geometry, and the surface termination groups in the MXene, which are impossible using top-down approaches [109,110,111,112]. However, compared to the top-down approach of MXene preparation, only a few studies have been performed with the bottom-up approach, perhaps due to the difficulty in the bottom-up approach to build atom-by-atom the MXene’s complex structure and multi-component atomic-thick layer. Moreover, the mechanisms of interaction between the layers are still not clear.

Hong et al. demonstrated the preparation of Mo_2_N MXene sheets by the chemical vapor deposition (CVD) method with NH_3_ as the source in the temperature range of 1080 °C [112]. Similarly, Ren et al., 2015 developed ultrathin α-Mo_2_CT_x_ crystals with excellent stability and defect free by CVD [113]. Buke et al. reported a detailed study on the effect of temperature, time, and copper layer thickness in the preparation of an MXene by the CVD method. In addition, they also found that Mo_2_C MXene crystals formed on the graphene sheets are thinner than those formed on the copper sheets [114]. In continuation to the CVD method, pulse laser deposition (PLD) and salt template methods were also investigated for the preparation of MXenes through a bottom-up approach [115,116].

### 3.2. Properties of Mxenes

MXenes have already proven their steadfast importance in multiple applications such as catalysis [117]; sensors (optical [118,119]; electrochemical [120,121,122,123,124,125,126]; surface-enhanced Raman scattering [127,128]; biomedical applications [129,130]; electromagnetic shielding [131,132,133]; and energy storage [134,135]. This success is significantly due to its unique properties, such as high young’s modulus [136,137], adjustable band gap [138], and thermal and electrical conductivity [139]. Comparison of fundamental properties of nanomaterials are shown in Table 2.

#### 3.2.1. Electrical Properties

Similar to the MAX phase, a pristine MXene is all-metallic. Thus, the research has been focused on enhancing the conductivity nature of MXenes by engineering their surface chemistry [140,141]. The prepared MXene using etchants leaves a surface termination group (–OH, –F, =O) which will bond to the metal atom in the MXene. Based on the experimental and theoretical studies, a few reports showed that some termination groups may affect the conductive property of the MXene [142,143,144] and electron mobility [145,146,147]. Yan et al. demonstrated an alkaline-treated MXene for humidity sensors, where they showed similar effects of –OH and –O termination groups in enhancing the electrochemical capacitance and sensor properties [107,148,149]. Pandey et al. investigated the electronic properties of M***_n_***C***_n_***_−1_O_2_ MXenes (M = Ti, W, Ta, Hf, Sc Mo, Nb, Cr, Zr, Y, Mn, V) [150], in which they found that oxygen-terminated MXenes are favorable for many applications.

**Table 2 molecules-28-04617-t002:** Comparison of fundamental properties of nanomaterials.

Nanomaterials	Conductivity (S/cm)	Surface Area (m^2^ g^−1^)	Biocompatibility	References
Graphene	2700	450	Biocompatible	[151,152]
Single-walled carbon nanotubes	10^2^ to 10^6^	600	Under debate	[153,154,155]
Multiwalled carbon nanotubes	10^3^ to 10^5^	122	Under debate	[153,154,155]
Hexagonal boron nitride	Insulator	150–550	Depends on the shape and size	[156,157]
MnO_2_	10^−5^ to 10^−6^	257.5	Biocompatible	[158,159,160]
MoS_2_	10^−4^	8.6	Biocompatible	[161,162,163]
MXene-Ti_3_C_2_	15100	93.6	Biocompatible	[164,165,166,167,168]

#### 3.2.2. Biocompatibility of MXenes

The principal mechanisms of nanomaterials’ toxicity could be one of the following: (1) damage to cells, (2) genotoxicity, (3) slow clearance in the renal pathway and accumulation in the organs, and (4) specific toxicity to the neural and reproductive system. Thus, an extensive biological safety evaluation is required to understand all the possible interactions between the nanomaterials and the physiological systems. MXenes, due to their high surface area and hydrophilicity, provide a suitable matrix for fabricating wearable biosensors [169]. Some preliminary results on MXene cytocompatibility and biocompatibility are already explored in the literature. Ti_3_C_2_ MXene is the most commonly investigated system in biomedical applications [166,170,171,172]. Liu et al. developed Ti_3_C_2_ MXene nanosheets for theranostics applications. They revealed that the MXene nanosheets passed in the bloodstream are excreted through human urine via the renal clearance pathway or accumulated in the tumor tissue due to the enhanced permeability and retention effect (EPR). In this study, the Ti_3_C_2_ nanosheets’ biosafety and biocompatibility are confirmed by the absence of significant weight loss and the absence of necrotic process. Han et al. investigated the in vivo biocompatibility of Ti_3_C_2_ MXene by administering the MXene to mice at elevated dosages (6.25, 12.5, 25, and 50 mg/kg), which resulted in excellent in vivo biocompatibility at 25 mg/kg [173]. In a similar study, Zong et al. developed a Ti_3_C_2_-GdW10 nanocomposite for cancer theranostics application. The in vivo biocompatibility studies concluded that up to 50 mg/kg of Ti_3_C_2_-GdW10 nanocomposite was biocompatible using female Kunming mice [174]. Similar results have been reported for systematic cytocompatibility and biocompatibility of tantalum carbide-based MXene Ta_4_C_3_ [171,175] and niobium carbide MXene Nb_2_C [176,177].

## 4. MXene-Based Electrochemical Sweat Sensors

Due to the fascinating physiochemical properties of MXenes and their composites, they have gained strong attention in biosensing with their excellent sensitivity, specificity, LOD, mechanical properties, and biocompatibility [136,144,178,179].

### 4.1. Glucose Sensing

Peng et al. developed a Pt/Ti_3_C_2_ MXene-based wearable, flexible non-enzymatic electrochemical biosensor for continuous monitoring of glucose in sweat [180]. Figure 4a–c illustrates the fabrication process of the sensor, microfluidic patch, and integration of the flexible sensor to the prepared patch. Figure 4d–f shows the conceptual scheme of the sensor for glucose detection, the cross-sectional view of the sensor on the skin, and the oxidation mechanism of the material and analyte. Figure 4g,h shows the electrochemical reaction mechanism of Pt/Ti_3_C_2_ coated on a glassy carbon electrode (GCE) and fabricated flexible sensor. This system offers a LOD of 29.15 μmol L^–1^ with a correlation coefficient of 0.9793.

In another work by Park et al., 2022, a butterfly-inspired hybrid epidermal biosensing (*bi*-HEB) patch, made with carbon Ti_3_C_2_T_x_ MXene-based nanocomposite, was developed for simultaneous monitoring of glucose and electrocardiograms (ECGs) in human subjects while performing indoor physical activities [73]. This system demonstrated excellent sensitivity of 100.85 µAmm^−1^ cm^−2^ within physiological levels (0.003−1.5 mm). Moreover, variations in pH and temperature from on-body sweat monitoring are calibrated. Figure 5a–c is a detailed schematic overview of *bi*-HEB patch fabrication. Feng et al. demonstrated a 3D porous Ti_3_C_2_T_x_/graphene/AuNP composite electrochemical biosensor for quick, responsive glucose detection. This system has a LOD of 2 μM in a concise time of fewer than 3 s with a sensitivity of 169.49 μA/(mM·cm^2^) in the range of 2 μM–0.4 mM concentration of glucose [181]. Hosseini et al. fabricated a Ti_3_C_2_/nickel-samarium-layered double hydroxide for enzyme-free real-time glucose detection. The glucose sensing was investigated with the DPV method, resulting in a LOD value of 0.24 μM and a linear range from 0.001 to 0.1 mM and 0.25–7.5 Mm [182].

Li et al. demonstrated a highly integrated wearable sensing paper (HIWSP) for electrochemical analysis of glucose and lactate from sweat [183]. This paper composition contains hydrophobic-layered protecting wax, a hydrophilic layer for sweat diffusion, and a Ti_3_C_2_T_x_ MXene/methylene blue composite. This sensor had a sensitivity of 2.4 nA μM^−1^ and 0.49 μA mM^−1^ for the simultaneous detection of glucose and lactate, respectively. In another work, Lei et al. fabricated a high-performing wearable biosensor for in vitro perspiration analysis [184]. This multifunctional biosensor is designed by incorporating Ti_3_C_2_T_x_ MXene and Persian blue composite for durable and sensitive monitoring of glucose and lactate in sweat. Figure 6a–i illustrates the real-time monitoring of wearable patches indicating pH, glucose, and lactate. Magesh et al. fabricated a palladium hydroxide integrated Ti_3_C_2_T_x_ MXene composite electrode to detect nicotine analyte in human sweat. This analysis was performed with the help of CV and amperometric studies and exhibited a LOD value of 27 nM with a sensitivity of 0.286 µA µM^−1^ cm^−2^ [185].

Ti_3_C_2_T_x_ MXene nanoflakes decorated ZnO tetrapods (ZnO TPs), a graphene oxide composite, and a skin-attachable enzymatic electrothermal glucose sensor was fabricated by Myndrul et al. [186]. This system showed a LOD of 17 μM with a broad linear detection range of 0.05–0.7 mM for detecting glucose from human sweat. Figure 7a–d shows stretchable electrodes at various strains, glucose sensing performance at applied strain (artificial sweat), skin-attachable sensor performance, and current density changes in the fabricated sensors on volunteer sweet consumption for glucose monitoring.

### 4.2. Cortisol Sensing

Park et al. developed a microfluidic integrated wearable impedimetric immune sensor based on Ti_3_C_2_T_x_ MXene incorporating laser-mediated porous graphene. This immune-sensing patch analyzes cortisol from the sweat by collecting sweat samples at one touch using a microfluidic channel network. Thus, wearable immune-sensing patches exhibit a dynamic range of 0.01–100 nM with a LOD of 88pM [188]. In another work, Laochai et al. fabricated Ti_3_C_2_ MXene/AuNPs/L-cysteine composited electrodes for real-time detection of cortisol from sweat [71]. Under optimal conditions, this fabricated sensor shows sensitivity with wide linearity of 5–180 ng mL^−1^ and a LOD value of 0.54 ng mL^−1^. The real-time analysis of cortisol from artificial sweat results in a 94.47–102% recovery value. The fabrication of cortisol immune sensors and immobilization steps are discussed with the help of Figure 8a,b.

### 4.3. Other Analytes

Zhi et al. developed a bioinspired directional moisture-wicking electronic skin for static health monitoring with the support of triboelectric energy harvesting. The device is made of heterogeneous fibrous polyvinylidene fluoride (PVDF) as a hydrophobic layer and polyacrylonitrile (PAN) as the hydrophilic layer with Ti_3_C_2_T_x_ MXene/CNT conductive ink [189]. Zhang et al. fabricated a wireless, battery-free, wearable, skin-interfaced electrochemical sensing patch for K^+^ monitoring from sweat. They used valinomycin as a K^+^ selective carrier specific to K^+^, enabling the selective detection of K^+^ by the membrane. Figure 9a–e demonstrates the conceptual operation mechanism of the fabricated device, electrode fabrication, sensor fabrication, and CV analysis of the material, followed by EIS analysis. This fabricated sensor measures the K^+^ ion concentration from human sweat with an excellent sensitivity of 63 mV/dec and a high linear detection range from 1 to 32 mM [77]. In another work, Hui et al. designed a flexible electrochemical heavy metal sensor for the non-invasive detection of Cu and Zn ions in human sweat with a material composition of Ti_3_C_2_T_x_/multi-walled carbon nanotubes (MWCNTs) in a layer-by-layer self-assembly [187]. The square wave anodic stripping voltammetric method (SWASV) was used for electrochemical analysis. The resulting LOD of Cu and Zn ions is 0.1 and 1.5 ppb. The response of SWASV for Cu (II) ion and the bending process of the fabricated electrode are shown in Figure 7e,f.

Cui et al. demonstrated a heterostructure Ti_3_C_2_ MXene/MoS_2_ composite for detecting ascorbic acid in human sweat. The study resulted in high sensitivity of 54.6 nA μM^−1^ and a LOD of 4.2 μM [190]. In another work, Saleh et al. fabricated inkjet-printed Ti_3_C_2_T_x_ MXene-based electrodes for cutaneous biosensing [74]. They demonstrated an exciting finding on Inkjet-printed MXene electrodes for Na^+^ ion detection and cytokine protein with 3.9 mV per decade by simple changes in the functionalization of the target analyte. These findings simplify the fabrication of wearable electronic platforms that enable multimodal biosensing. Qiao et al. developed a Ti_3_C_2_ MXene@TiO_2_ (anatase/rutile) ternary heterostructured electrode to detect phosphoprotein in sweat. They showed a LOD of 1.52 μM with a broad linear range from 0.01 to 1 mg/mL [191]. Chen et al. developed a fluoroalkyl functionalized F–Ti_3_C_2_T_x_/polyaniline (PANI) superhydrophobic skin-attachable and wearable electrochemical pH biosensor for real-time perspiration analysis [192]. Thus, the schematic view of the sensing device setup, pH analysis of sweat in males/females, and comparison of measured values from the fabricated patch with other equipment are discussed in detail in Figure 10a–d.

## 5. Summary and Outlook

In recent years, more attention has been paid to MXenes due to their outstanding characteristic features such as high conductivity, surface area, oxidation and redox capacity, biocompatibility, and significant electrocatalytic and electrochemical properties. Moreover, MXenes also hold great potential as conductive substrates for various electrode-based devices, especially for specific-patterned electrodes. This paves the way for the fabrication of highly integrated electrochemical sensing gadgets such as wearable sensors for non-invasive monitoring of biomarkers in the body fluid (sweat) and miniaturized electronic devices for practical standards. However, for the faster translation of MXene-based electrochemical sensors from lab to market, we propose the following research directions in the near future:

(i) Device fabrication with precise nano-/micro-patterned structures are essential to enhance the performance of wearable sensors, and their shortcomings cannot be ignored [193].

(ii) It is challenging to synthesize MXenes on a large scale due to their atomic-thick layer design. To overcome this, future research should address the industrial-scale nanostructured design of MXenes and cost-effective technologies [194,195].

(iii) The malleable structure is crucial in changing the conductivity and internal resistance of the MXene. Hence future research should focus on the enlargement of interlayer distance in MXenes.

(iv) The problems with the flexibility of MXenes can be solved by performing focused research on MXene-polymer nanocomposites using biocompatible polymers [196].

(v) MXene-based biosensors strongly rely on nanohybrid biocompatibility. Thus, there should be focused research on the surface chemistry of MXenes to solve the problems based on the affinity and stability of biomolecules on MXene surfaces [197].

(vi) In the case of wearable sensors, MXene nanomaterial is oxidized when continuously in contact with air. This reduces the conductivity and affects the sensing ability. However, the external polymer coating to prevent oxidation in the MXene affects the breathability and comfort of the wearable biosensors. Thus, an in-depth understanding is needed to design sensors that could maintain the conductivity of the MXene without losing the convenience of the user [169].

## Figures and Tables

**Figure 1 molecules-28-04617-f001:**
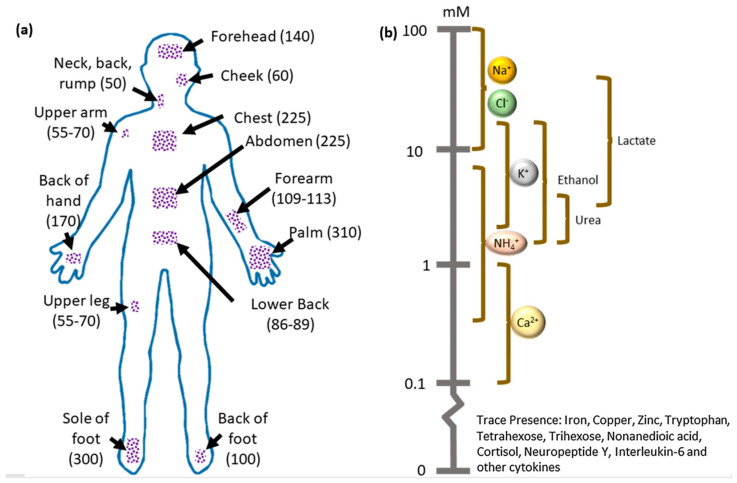
(**a**) Schematic representation of the average density of sweat glands in the different parts of the human body(glands/cm^2^). (**b**) Approximate range of analytes’ concentrations present in the sweat. Reproduced with permission from Ref. [60]. Copyright 2019 Elsevier.

**Figure 2 molecules-28-04617-f002:**
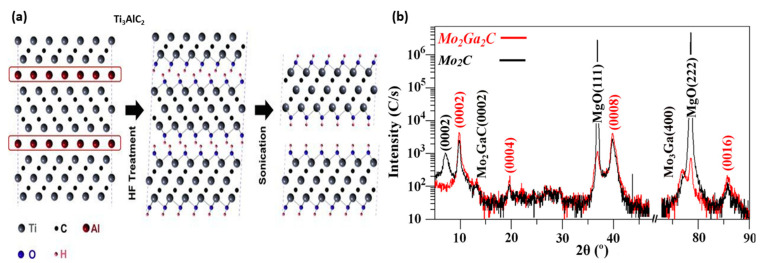
(**a**) Schematic illustration of the synthesis process of Ti_3_AlC_2_. Reproduced with permission from Ref. [85] Copyright 2011 John Wiley and Sons. (**b**) X-ray diffraction pattern of Mo_2_Ga_2_C film before and after HF etching. Reproduced with permission from Ref. [86]. Copyright 2015 Elsevier.

**Figure 3 molecules-28-04617-f003:**
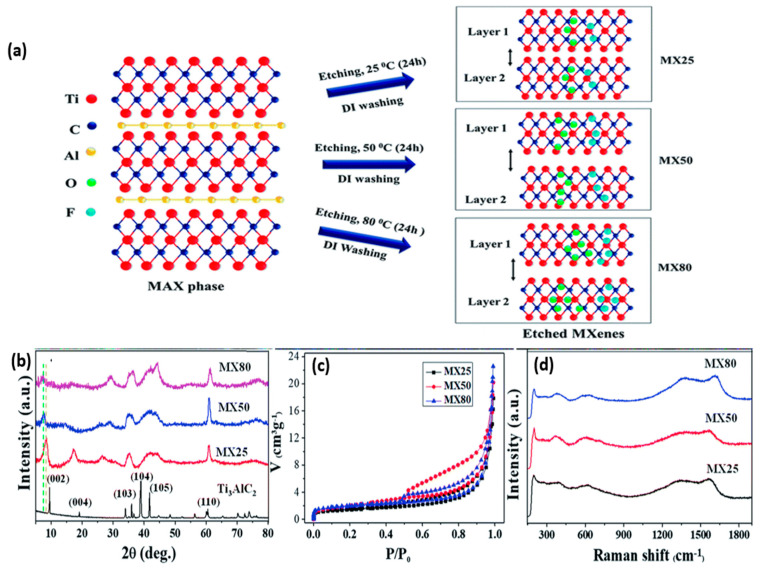
(**a**) Schematic representation of the etching mechanism at elevated temperature and their high exfoliation. (**b**) XRD pattern of exfoliated material at different temperatures and reduced intensity at high–temperature etching of MX80 indicates deterioration of crystallinity. (**c**) BET isotherm studies confirm the increase in surface area with high–temperature etching. (**d**) Raman spectra of exfoliated MXene at different temperatures were high–intensity peaks observed for MX80, indicating the defects introduced at high–temperature etching. Reproduced from Ref. [99] with permission from the Royal Society of Chemistry.

**Figure 4 molecules-28-04617-f004:**
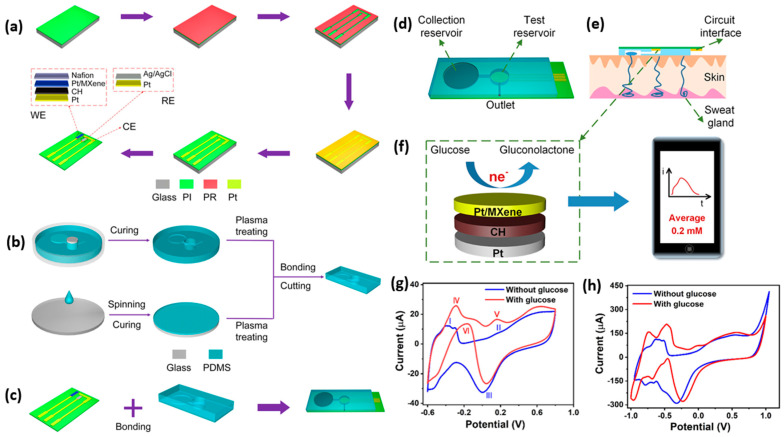
(**a**) Schematics show the flexible sensor’s fabrication process. (**b**) The preparation process of the microfluidic patch. (**c**) Integration of MXene–based sensor to microfluidic patch. (**d**) Schematic overview of the proposed flexible wearable sensor. (**e**) Cross–sectional view of the sensor on the skin. (**f**) The electrochemical oxidation reaction of glucose on the MXene. (**g**) CV response of Pt/Ti_3_C_2_ MXene coated on GCE. (**h**) CV response of Pt/Ti_3_C_2_–fabricated flexible wearable sensor. The figure is reproduced with permission from [180]. The copyright year is 2023 American Chemical Society.

**Figure 5 molecules-28-04617-f005:**
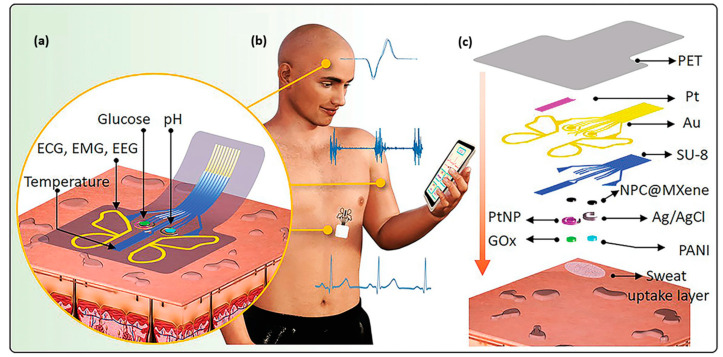
(**a**) Schematic illustration of layer-by-layer assembled multiplexed electrochemical sensor mounted on the skin containing various analyte-templated patterns designed on thin polyethylene terephthalate (PET) substrate. (**b**) *bi*-HEB patch mounted at the chest to monitor sweat glucose, temperature, and pH simultaneously. (**c**) Schematic representation of layer-by-layer fabrication process of the *bi*-HEB patch from top to bottom. Reproduced with permission from Ref. [73]. Copyright 2022 John Wiley and Sons.

**Figure 6 molecules-28-04617-f006:**
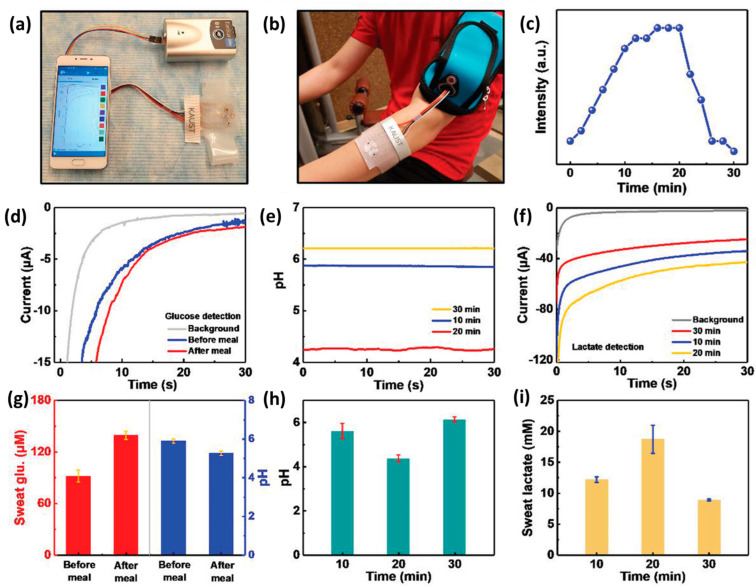
(**a**) Photograph of wearable sweat monitoring patch connected to a portable electrochemical analyzer that supplies power and can wirelessly communicate the body’s status with mobile phones via Bluetooth. (**b**) Skin–attached wearable sweat sensor. (**c**) Graphical data of on–body test cycling resistance profile. (**d**) Chronoamperometric response of pH changes and glucose sensor before and after a meal. (**e**) pH response at different times during exercise. (**f**) The electrochemical response of lactate sensor at different times during exercise. (**g**) Comparison of pH level and glucose after a meal with three different sensors. (**h**) Comparison of pH levels at different times during exercise. (**i**) Comparison of lactate sensor responses at different exercise times with different lactate sensors. Reproduced with permission from Ref. [184]. Copyright 2019 John Wiley and Sons.

**Figure 7 molecules-28-04617-f007:**
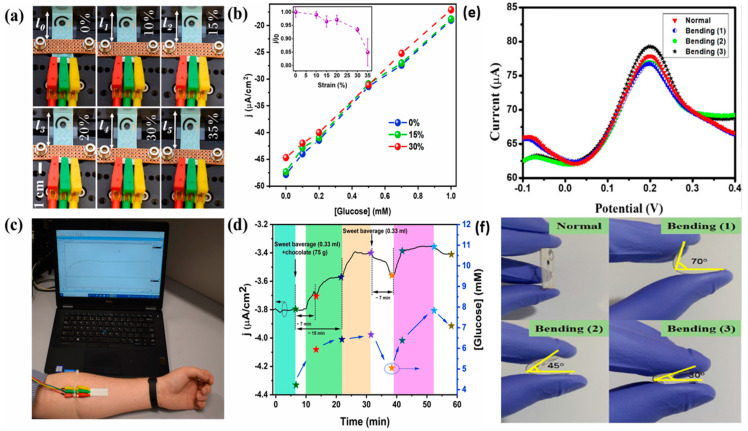
(**a**) Picture of flexible MXene/ZnO TPs/Gox composite electrode at strain range of 0–35%. (**b**) Working performance of fabricated sensor at applied strains (electrode current density *vs* applied strains) graph. (**c**) Photograph of skin–attachable sensor functioning for monitoring of glucose in sweat. (**d**) Graph of current density changes in composite material under sweet consumption for glucose detection. Reprinted with permission from Ref. [186]. (**e**) SWASV response of composite material–based fabricated electrode for 300 ppb Cu (II) ion on normal and bending mode. (**f**) Photographs of bending processes of the fabricated electrode. Reproduced with permission from Ref. [187]. Copyright 2020, American Chemical Society.

**Figure 8 molecules-28-04617-f008:**
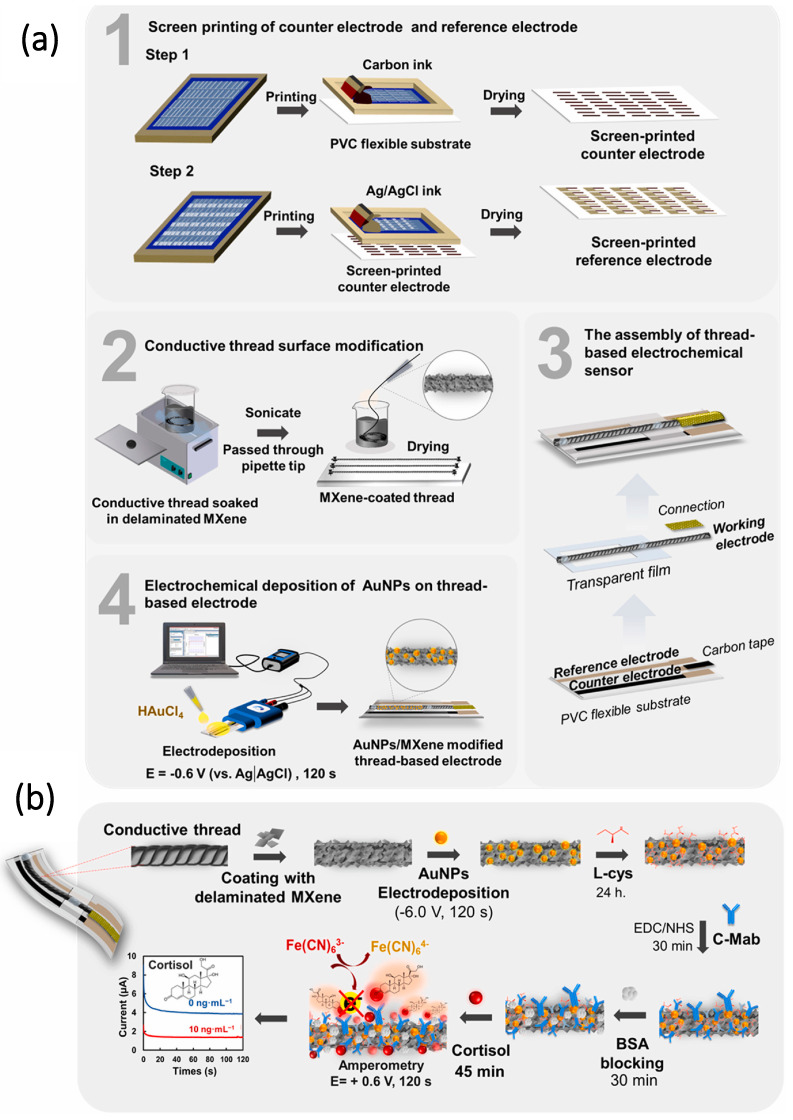
(**a**) Schematic illustration of the fabrication of counter and reference electrodes, surface modification, and electrochemical deposition of gold nanoparticles on the surface of the electrode. (**b**) Immobilization process. Reproduced with permission from Ref. [71]. Copyright 2022 Elsevier.

**Figure 9 molecules-28-04617-f009:**
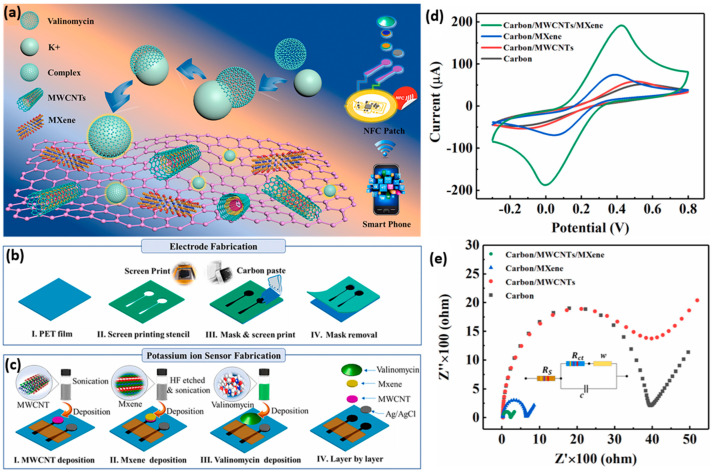
(**a**) Schematic illustration of the operating mechanism of wearable, wireless, battery–free integrated electrochemical sensing patch (valinomycin is a selective K^+^ carrier and naturally has specific permeability to K^+^). (**b**) Steps involved in electrode fabrication. (**c**) Sensor fabrication process. (**d**) CV study bare carbon, carbon/MWCNTs, carbon/MXene, and carbon/MWCNTs/MXene. (**e**) EIS Nyquist analysis. Reproduced with the permission form. Ref. [77]. Copyright 2020 Elsevier.

**Figure 10 molecules-28-04617-f010:**
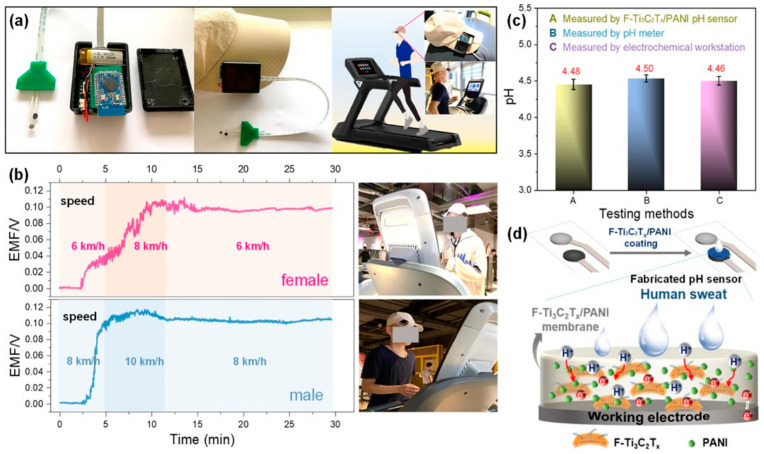
(**a**) Digital photograph of wearable pH sensor with a lithium-ion battery, mini–type potentiometer, and F–Ti_3_C_2_T_x_/PANI material. (**b**) Continuously monitored real–time pH in human male and female volunteers. (**c**) Comparison of F–Ti_3_C_2_T_x_/PANI sensor with ex situ electrochemical workstation and pH meter results. (**d**) Schematic illustration of F–Ti_3_C_2_T_x_/PANI electrochemical behavior. Reproduced with permission from Ref. [192]. Copyright 2022 American Chemical Society.

**Table 1 molecules-28-04617-t001:** Key analytes present in human sweat and associated with health issues.

Analytes	Health Condition	Refs.
Iron	Sports anemia	[25]
Glucose	Key analyte for diabetic conditions	[61,62]
Lactate	Analytes accumulated during a transition from aerobic to anaerobic conditions.	[63,64]
Uric acid	Gout, Renal dysfunction	[65]
Zinc (Zn^2+^)	Immune system-induced muscle damage	[66,67]
Copper (Cu^2+^)	Rheumatoid arthritis, Cirrhosis of liver	[67,68]
Interleukin 6	Proinflammatory cytokines, markers for certain cancers, and inflammation	[69,70]
Cortisol	Stress	[71]
Neuropeptide Y	Stress	[72]
pH	Wound healing, Skin diseases—pathogenic	[73]
Sodium (Na^+^)	Dehydration, Electrolyte imbalance, Hyponatremia	[74,75]
Cl^−^	Cystic fibrosis, Dehydration	[76]
Potassium (K^+^)	Hypokalaemia, Muscle cramps	[77]
Calcium (Ca^2+^)	Renal failure, Acid-base balance disorder, Myeloma	[78]
Ammonium (NH_4_^+^)	The shift from aerobic to anaerobic conditions	[79,80]

## Data Availability

All the data and materials that support the results or analyses presented in their paper are freely available upon request.

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
