# Peer review of "Recent Advances in Two-Dimensional MXene-Based Electrochemical Biosensors for Sweat Analysis"

_molecules, 2023, doi:10.3390/molecules28124617_

Round 1

Reviewer 1 Report

This manuscript presents the recent progress made in 2D MXene-based bio-electrochemical sensors such as wearable, implantable, and microfluidic sensors, and their applications in disease diagnosis and developing point-of-care sensing platforms, such as sweat sensing, glucose sensing, cortisol sensors, etc. Finally, the paper discusses the challenges and limitations of MXene as a material of choice in bioelectrochemical sensors and future perspectives on this exciting material for sweat-sensing applications. The author review and discuss various synthesis methods for MXenes, their surface chemistry, and functionalization strategies in sensor applications. The advantages and importance of sweat analytes in continuously monitoring human physiology and disease conditions. Finally, the author also discuss various MXene-based wearable biosensors for vital analytes in sweat. Most importantly, this review sheds light on future scopes and recommendations for researchers working in MXene for electrochemical biosensor applications. In order to improve the manuscript, the following issues should be considered.

1. As the article says, the composition of human sweat is complex and diverse, so whether the selectivity of MXene material, the ability to bind specifically to the objects to be tested, is confirmed.

2. In the third part, it is mentioned that MXene has better biocompatibility than carbon materials. Please use relevant experiments or data to prove and explain what the principle of biocompatibility is.

3. In Section 3.2 describes only the high conductivity of this material, but not its biocompatibility.

4. The third part describes that the cost is low, but at the end of the article presents it is high. Theyre contradictory.

5. Please add in the introduction what are the important analytes of sweat compared to other body fluids and how these analytes are physiologically related to the human body.

6. Please specify the mechanism by which MXene has superior biocompatibility and conductivity compared to conventional metal nanoparticle-based and graphene derivative-based biosensing.

7. In “Synthesis Strategies of MXene”, the author provided a detailed introduction to the preparation methods and difficulties of the Top down approach, but there was relatively little introduction to the Bottom up approach. The overall presentation presented the technical characteristics of the previous method, which was complex and difficult. However, there was no recognition given to the relatively simple method of the latter, making it difficult to distinguish where the author wanted to highlight.

8. Please update references style.

Author Response

Response to reviewers’ comments

The authors would like to thank the reviewers for their comments, suggestions, and comprehensive evaluation of our manuscript. The critical comments and suggestions encourage the authors to improve the quality of the manuscript. We also believe that the manuscript is now more concise, clear, and relevant to the readers. In the revised manuscript, the changes are highlighted in yellow colour. Here are the point-by-point responses to the reviewer’s comments.

Reviewer 1

  1. As the article says, the composition of human sweat is complex and diverse, so whether the selectivity of MXene material, the ability to bind specifically to the objects to be tested, is confirmed.

MXene’s binding ability is based its characteristic features like hydrophilicity and high surface area. There is no literature that support the specific binding of analytes (present in sweat) to the MXene. However, the specificity is typically incorporated through immobilization of bio-recognition elements like antibodies, aptamers and many other biorecognition moieties, which are specific to the analytes of our interests. We have already discussed them and referenced them in our manuscript. Please see in references 178 to 190

  1. In the third part, it is mentioned that MXene has better biocompatibility than carbon materials. Please use relevant experiments or data to prove and explain what the principle of biocompatibility is.

We thank the reviewer for this valuable comment. As per the reviewer’s suggestion, we have included a separate section to discuss about the biocompatibility of MXene. Please see in section 3.2.2.

  1. In Section 3.2 describes only the high conductivity of this material, but not its biocompatibility.

As per the reviewers’ suggestion we have included sperate portion to discuss about the biocompatibility of MXene. Please see in section 3.2.2.

  1. The third part describes that the cost is low, but at the end of the article presents it is high. They’re contradictory.

We thank the reviewer for pointing this contradictory sentence. We have mentioned “high cost” in summary based on expensive precursors and its high demand. However, in case of electrochemical applications, due to excellent conductivity of MXene and unique characteristics like high surface area, hydrophilicity, we need very small amount of them for electrochemical applications when compared to other conductive materials to have the same conductivity. Thus, we claim them to be low cost for electrochemical applications. To avoid the contradictory statement, we have removed the word “high cost” from the summary portion.  Please see in line # 418.

  1. Please add in the introduction what are the important analytes of sweat compared to other body fluids and how these analytes are physiologically related to the human body.

As per the reviewer’s suggestions, we have added the important analytes of sweat in the introduction portion and highlighted. Please see in line # 56 to 60. The physiological relation of sweat analytes and their associated health issues are discussed in section 2 and table 1.

  1. Please specify the mechanism by which MXene has superior biocompatibility and conductivity compared to conventional metal nanoparticle-based and graphene derivative-based biosensing.

In general, the conductivity and biocompatibility of MXene are highly based on its properties like, high surface area, hydrophilicity and tunable surface terminations, which are discussed in the section 3.2.1 and 3.2.2 with support of literatures. Please see in line 231 to 270

  1. In “Synthesis Strategies of MXene”, the author provided a detailed introduction to the preparation methods and difficulties of the Top-down approach, but there was relatively little introduction to the Bottom-up approach. The overall presentation presented the technical characteristics of the previous method, which was complex and difficult. However, there was no recognition given to the relatively simple method of the latter, making it difficult to distinguish where the author wanted to highlight.

Please see in line 216 to 218 which describes the advantage of bottom-up method and limitation in preparing and understanding the bottom-up approach.

  1. Please update references style.

As per the reviewer’s suggestion. We have updated the reference style.

Reviewer 2 Report

The review article “Recent advances in 2D MXene-based electrochemical biosensors for sweat analysis” reports the advances of MXenes materials in electrochemical field. The review is well written, I have minor considerations, such as:

1.     Keywords must be different from the title to get more visibility to the work;

2.     The graphical abstract may suffer improvements, such as color, quality of electrode designs, etc…

3.     In general, I missed authors explore other MXenes such as MoC2Tx, NbC2Tx…

4.     Line 187, NH4+ is wrong written

5  5. Authors should add more references of 2022 and 2023.

Author Response

Response to reviewers’ comments

The authors would like to thank the reviewers for their comments, suggestions, and comprehensive evaluation of our manuscript. The critical comments and suggestions encourage the authors to improve the quality of the manuscript. We also believe that the manuscript is now more concise, clear, and relevant to the readers. In the revised manuscript, the changes are highlighted in yellow colour. Here are the point-by-point responses to the reviewer’s comments.

Reviewer 2

  1. Keywords must be different from the title to get more visibility to the work;

We have modified the keywords as per the reviewer’s comments. Please see line # 33

  1. The graphical abstract may suffer improvements, such as colour, quality of electrode designs, etc…

As per the reviewer’s suggestion, graphical abstract is modified.

  1. In general, I missed authors explore other MXenes such as MoC2Tx, NbC2Tx…

There are no literatures on MoC2Tx, NbC2Tx, TaCTx, V2CTx, and W2CTx MXene in sweat biosensor particularly.

  1. Line 187, NH4+ is wrong written

We thank reviewer for pointing this out. We corrected the molecular formula, please see in line # 196

  1. Authors should add more references of 2022 and 2023.

As per reviewer’s suggestion. We have added more references

Please see in reference number [31, 105. 116, 122, 123, 124, 136, 169, 195]

Reviewer 3 Report

The authors have written a comprehensive review on the recent advances in 2D MXene-based electrochemical biosensors for sweat analysis. After carefully reading the whole manuscript, I do recommend this paper to be published after some modifications. The authors should address the following issues.

1.      The graphical abstract can be improved. There are a lot of empty white spaces within the figure.

2.      The first two paragraphs of the introduction lack literature support. It has only 8 references. Kindly include a few more relevant literature.

3.      The last part of the introduction should emphasize the importance and highlights of the present manuscript.

4.      In section 3.1.2.2. what happens to the MXenes in terms of their electrochemical activity, stability, etc, if we delaminate the with intercalating agents?

5.      The properties of MXene could be a little more elaborate and detailed.

6.      What is (F) in Fig. 4 caption, line no. 5? And there is no detail about (h) in the figure caption.

7.      Even though the summary and outlook appear informative and thoughtful, it lacks connectivity. The authors should coherently rewrite the summary part.

8.      A detailed list of abbreviations needs to be included after the conclusion.

9.      Overall, the manuscript could be a little more in detail since it has reviewed a collection of 130 pieces of literature. It appears that the manuscript has more figures compared to the text written in it. I suggest the authors improve the discussions in more detail in order to make this review a comprehensive one.

Overall the language used in the manuscript is good. However, it should be proofread for typos and syntax errors

Author Response

Response to reviewers’ comments

Reviewer 3

  1. The graphical abstract can be improved. There are a lot of empty white spaces within the figure.

As per the reviewer’s suggestion, graphical abstract is modified.

  1. The first two paragraphs of the introduction lack literature support. It has only 8 references. Kindly include a few more relevant literatures.

We have added few more references to support the introduction as per the reviewer’s suggestion. Newly added references are highlighted in yellow colour.

  1. The last part of the introduction should emphasize the importance and highlights of the present manuscript.

We thank reviewers for the valuable suggestions. The last part of the introduction emphasizing the importance and highlights of the manuscript are highlighted in yellow colour. Please see in line 85 to 90.

  1. In section 3.1.2.2. what happens to the MXenes in terms of their electrochemical activity, stability, etc, if we delaminate the with intercalating agents?

While delamination with intercalating agents it reduces the interlayer spacing and weaken the interaction in MXene layer to from individual nanosheets. This greatly improves the surface area and abundant surface terminal groups which directly associated with MXene conductive properties.

We thank reviewers for the question. We reframed the words in the section 3.1.2.2

[Old: reduce the interlayer spacing and

weaken the interaction between the MXene layers for productive delamination]

[Revised: reduce the interlayer spacing and weaken the interaction between the MXene layer and facilitate to form individual nanosheet]

 please see in line # 163 to 166

  1. The properties of MXene could be a little more elaborate and detailed.

As per the reviewer’s suggestions, we have elaborated the properties section by discussing the biocompatibility and electrical conductivity of MXene and added table on fundamental properties of various nanomaterials.

  1. What is (F) in Fig. 4 caption, line no. 5? And there is no detail about (h) in the figure caption.

We thank reviewers for pointing the error and apologies for the error. We have corrected the figure caption. Please see in line # 279.

  1. Even though the summary and outlook appear informative and thoughtful, it lacks connectivity. The authors should coherently rewrite the summary part.

We thank reviewer for this valuable suggestion. As per the reviewer’s comment, we have modified the summary portion. Please see in line 405 to 431.

  1. A detailed list of abbreviations needs to be included after the conclusion.

As per the reviewer’s suggestion, we have added the detailed abbreviation after the conclusion. Please see in line 432.

  1. Overall, the manuscript could be a little more in detail since it has reviewed a collection of 130 pieces of literature. It appears that the manuscript has more figures compared to the text written in it. I suggest the authors improve the discussions in more detail in order to make this review a comprehensive one.

As per the reviewer’s suggestion. We have added new sections, tables and references to support the literature.

Newly added sections 3.2.1 and 3.2.2

Newly added table please see in line 244.

Comments on the Quality of English Language

Overall, the language used in the manuscript is good. However, it should be proofread for typos and syntax errors

Round 2

Reviewer 1 Report

No more comments.

Author Response

Thank you for your vaulable comments. 

Reviewer 3 Report

After the revision, the manuscript's quality significantly improved and I recommend the Editor to consider this manuscript for publication in Molecules. Here are a few corrections that should be carried out before its acceptance. 

1. There is a numbering error in 3.2 (i.e. 3.2.1 is missing and  3.2.2 appears twice) 

2. Fig 3 (a) caption. remove bold (unbold)

3. The abbreviations should be in the alphabetical order 

4. Abbreviation for NH4 the '+' sign is missing

5. milliliter, pico molar, micro molar, nano molar are all one word. Combined it

6. Moreover, I request the authors to proofread their manuscript thoroughly 

It is good

Author Response

Reviewer 3

  1. There is a numbering error in 3.2 (i.e., 3.2.1 is missing and 3.2.2 appears twice) 

We thank reviewer for pointing this out. We corrected the numbering, please see in line # 261 and 275.

  1. Fig 3 (a) caption. remove bold (unbold)

We thank reviewer for pointing this out. We have corrected the figure 3(a) caption, please see in line # 201

  1. The abbreviations should be in the alphabetical order 

As per the reviewer’s suggestions, Abbreviations are modified. Please see in line # 479 to 513

  1. Abbreviation for NH4 the '+' sign is missing

We thank reviewer for pointing this mistake. We corrected the molecular formula of ammonium. Please see in line # 501

  1. milliliter, pico molar, micro molar, nano molar are all one word. Combined it

We thank reviewer for pointing this out and apologies for the mistake. We have corrected the words please see in line # 493, 494, 495, 503 and 508

  1. Moreover, I request the authors to proofread their manuscript thoroughly 

As per the reviewer’s suggestions, we have revised the manuscript and corrected the errors.
